

# Plant host and drought shape the root associated fungal microbiota in rice

Beatriz Andreo-Jimenez[1,2], Philippe Vandenkoornhuyse[3], Amandine Lê Van[3], Arvid Heutinck[1], Marie Duhamel[3,4], Niteen Kadam[5], Krishna Jagadish[5,6], Carolien Ruyter-Spira[1] and Harro Bouwmeester[1,7]

[1] Laboratory of Plant Physiology, Wageningen University, Wageningen, Netherlands
[2] Biointeractions & Plant Health Business Unit, Wageningen University & Research, Wageningen, Netherlands
[3] EcoBio, Université Rennes I, Rennes, France
[4] IBL Plant Sciences and Natural Products, Leiden University, Leiden, Netherlands
[5] International Rice Research Institute, Los Baños, Philippines
[6] Department of Agronomy, Kansas State University, Manhattan, KS, United States of America
[7] Plant Hormone Biology group, Swammerdam Institute for Life Sciences, University of Amsterdam, Amsterdam, Netherlands

## ABSTRACT

**Background and Aim**. Water is an increasingly scarce resource while some crops, such as paddy rice, require large amounts of water to maintain grain production. A better understanding of rice drought adaptation and tolerance mechanisms could help to reduce this problem. There is evidence of a possible role of root-associated fungi in drought adaptation. Here, we analyzed the endospheric fungal microbiota composition in rice and its relation to plant genotype and drought.

**Methods**. Fifteen rice genotypes (*Oryza sativa* ssp. indica) were grown in the field, under well-watered conditions or exposed to a drought period during flowering. The effect of genotype and treatment on the root fungal microbiota composition was analyzed by 18S ribosomal DNA high throughput sequencing. Grain yield was determined after plant maturation.

**Results**. There was a host genotype effect on the fungal community composition. Drought altered the composition of the root-associated fungal community and increased fungal biodiversity. The majority of OTUs identified belonged to the Pezizomycotina subphylum and 37 of these significantly correlated with a higher plant yield under drought, one of them being assigned to *Arthrinium phaeospermum*.

**Conclusion**. This study shows that both plant genotype and drought affect the root-associated fungal community in rice and that some fungi correlate with improved drought tolerance. This work opens new opportunities for basic research on the understanding of how the host affects microbiota recruitment as well as the possible use of specific fungi to improve drought tolerance in rice.

Corresponding author
Harro Bouwmeester,
h.j.bouwmeester@uva.nl

## INTRODUCTION

Climate change is one of the main driving forces affecting the environment. The resulting higher temperatures act to reinforce the effect of drought (*Trenberth et al., 2014*). Drought

periods are one of the main causes of grain yield losses in crops worldwide, especially in drought sensitive crops such as rice (*Oryza sativa*), the second most produced and consumed crop in the world. To ensure high productivity, rice requires well-watered conditions and almost half of the fresh water used for crop production worldwide is consumed by rice (*Barker et al., 2000*). As such, improving yield under drought is a major goal in rice breeding.

The root system is in direct contact with the soil, from which the plant absorbs water, and thus root traits are among the critical factors that can potentially ensure good yields under drought stress. Besides the root system and the plant itself, the interaction between plant root and symbiotic microorganisms forming the root microbiota is now considered a major factor in plant performance. These microorganisms may allow the plant to buffer the environmental constraints (*Vandenkoornhuyse et al., 2015*) and mitigate or suppress soil borne diseases (*Kwak et al., 2018*). Root colonizers include arbuscular mycorrhizal fungi (Glomeromycota) (*Augé, 2001*; *Smith & Read, 2008*; *Singh, 2011*), non-mycorrhizal fungal endophytes from the Ascomycota (such as the Pezizomycotina) and, to a lesser extent, the Basidiomycota. Root-associated fungi have repeatedly been reported to play a role in plant tolerance to stresses (e.g., *Selosse, Baudoin & Vandenkoornhuyse, 2004*; *Rodriguez et al., 2009*). Fungal endophytes have a broad host range and colonize the shoots, roots and rhizomes of their hosts (*Rodriguez et al., 2009*). They can increase plant biomass (*Ernst, Mendgen & Wirsel, 2003*; *Redman et al., 2011*; *Jogawat et al., 2013*) and improve tolerance to biotic (*Mejía et al., 2008*; *Maciá-Vicente et al., 2008*; *Chadha et al., 2015*) and abiotic stresses (*Hubbard, Germida & Vujanovic, 2014*; *Yang, Ma & Dai, 2014*; *Azad & Kaminskyj, 2015*).

The root fungal microbiota community is not static and changes with environmental factors. Pesticide application, for example, increases the richness of the AM fungal community composition in roots (*Vandenkoornhuyse et al., 2003*). In contrast, farming practices such as tillage and ploughing are known to decrease species richness of AM fungi in agricultural soils (e.g., *Verbruggen & Kiers, 2010*). Monocropping and conventional paddy cultivation also reduce the AMF diversity and colonization in rice and favor the presence of fungal pathogens (*Lumini et al., 2010*; *Esmaeili Taheri, Hamel & Gan, 2016*). In traditionally flooded rice fields, root associated fungal species in the Pleosporales and Eurotiales were less abundant than in roots of plants grown in upland fields (*Pili et al., 2015*).

Despite its reported role in plant fitness, the importance of plant colonizing fungal microbiota is underestimated, both in terms of diversity and functionality (*Lê Van et al., 2017*). Plants cannot be regarded as standalone entities but rather as holobionts comprised of the plant and its associated microbiota where the microbial community provides additional functions to help the cope with environmental changes and stresses (*Vandenkoornhuyse et al., 2015*). In this conceptual framework, recruitment by the host of micro-organisms when faced with constraints could explain microbiota heterogeneity on the same host in different developmental stage or under changing environmental conditions. If the host indeed exerts control on the recruitment of microorganisms, it is likely that genetic variation for this trait exists. Indeed, the phyllosphere bacterial

community in *Arabidopsis thaliana* (*Horton et al., 2014*) and wild mustard (*Wagner et al., 2016*) but also the barley root bacterial microbiota (*Bulgarelli et al., 2015*) are to some extent host-dependent suggesting that plants indeed exert control on microbial community recruitment from the microorganisms present in the soil. For the present study, we therefore hypothesized that changes that occur within the fungal microbiota community composition when plants experience an environmental constraint are (partially) determined by the plant genotype. To address this hypothesis, we analyzed the effect of drought on changes in the root associated fungal microbiota of a range of different rice cultivars and whether these changes may play a role in protecting rice against drought.

## MATERIALS & METHODS

### Plant Materials

Fifteen rice cultivars (*Oryza sativa* ssp. indica) from the International Rice Research Institute (IRRI, Los Baños, Philippines) were used in our study. Ten out of the 15 cultivars were selected to maximize the genetic variation using the SNP information available from a published study (*Zhao et al., 2011*). The five additional cultivars were selected based on their drought tolerance phenotype, and their information is available in IRGCIS database: http://www.irgcis.irri.org:81/grc/SearchData.htm (Table S3).

### Field site and growing conditions

All rice plants were grown at IRRI facilities from December 2012 to March 2013. The upland field (used to grow rice under non-flooded conditions) was located at 14°08′50.4″N 121°15′52.1″E. There were 45 field blocks (three per cultivar) (0.8 × 2.5 m) and each block included 48 plants. The three replicates of each cultivar were analyzed separately. The minimum distance between blocks was three meters. An additional 45 blocks were used for the drought treatment, so in total there were 90 blocks. The soil was a mix of clay (36%), sand (22%) and silt (41%). The plot design was randomized through the field site. Plants were grown in waterlogged conditions until 50% of the plants reached the flowering stage. Then a drought treatment was imposed on half of the replicates by withholding irrigation. After 12 days of drought, the stressed plots reached—46 KPa of soil water potential, while the control plot was saturated with water (100% of soil field capacity). There were no rain events during the stress imposition period. Since the plots were maintained under upland conditions with higher sand and silt and during the hotter tropical months of the Philippines, the targeted stress levels were reached in a relatively short duration of 12 days. Then, three soil cores of 10 × 70 cm diameter x length were collected from the center of the plots of the cultivars, pooled together (per block, so giving three replicate samples per genotype) and stored in plastic bags at 4 °C until further use. To remove all soil particles, roots isolated from the soil cores were carefully washed with tap water frozen in liquid $N_2$ and stored at −80 °C until use.

### DNA isolation and sequencing

Each root sample was grinded to powder with a mortar and pestle using liquid nitrogen, and DNA was extracted from 60–80 mg of plant material with the

DNeasy Plant Mini Kit (Qiagen) following the manufacturers protocol. From the extracted DNA, we amplified a fragment of the 18S SSU rRNA gene using general fungal primers (NS22: 5′-AATTAAGCAGACAAATCACT-3′and SSU0817: 5′-TTAGCATGGAATAATRRAATAGGA-3′) (*Borneman & Hartin, 2000*) and the following thermocycler conditions during the PCR: 94 °C for 3 min; 35 cycles of 94 °C for 45 s, 59 °C for 45 s (−0.1 °C/cycle), 72 °C for 1 min; and 72 °C for 10 min. Primers were modified to allow the amplicon multiplexing for the sequence production process. Primer modifications and PCR conditions followed *Lê Van et al., 2017*. To analyze the entire diversity of the fungal community that is associated with roots, including Chytridiomycota, early diverging lineages related to the former Zygomycota (onwards called Zygomycota) and Glomeromycota (*Sanders, Clapp & Wiemken, 1996*), SSU rRNA gene primers have been shown to successfully amplify unknown fungal species or groups (*Vandenkoornhuyse et al., 2002*; *Quast et al., 2013*; *Lê Van et al., 2017*).

PCR amplicons were purified with AMPure XP beads (Beckman Coulter). Amplicon size was verified with the Agilent High Sensitivity DNA kit (Agilent Technologies), and the concentration measured using the Quant-ITTMPicoGreen® dsDNA Assay kit (Invitrogen). Finally, the purified 560 bp amplicons were all diluted to similar concentration ($10^9$ copies), pooled and sequenced (454 GS FLX+ version Titanium; Roche), following the manufacturer's guidelines.

All the PCRs were performed twice and sequenced separately. These true replicates were used within our trimming strategy.

## Sequence data trimming and clustering

After demultiplexing, sequences were filtered to remove reads containing homopolymers longer than 6 nucleotides, undetermined nucleotides, anomalous length and differences (one or more) in the primer. Quality trimming and filtering of amplicons, OTU identification, and taxonomic assignments were carried out with a combination of amplicon data analysis tools and in-house Python scripts as described in *Lê Van et al., 2017*. In more detail, the sequences which passed all the filters were clustered using DNAClust (*Ghodsi, Liu & Pop, 2011*). Operational Taxonomic Units (OTUs) were generated out of a minimum of two 100% identical sequences that appeared independently in the different replicates. After these steps, filtering of chimeric sequences was performed using the 'chimeric.uchime' tool within Mothur (v1.31.0, *Schloss et al., 2009*). The trimming and clustering pipeline used was the same as used in previous studies (e.g., *Ben Maamar et al., 2015*; *Lê Van et al., 2017*). The affiliation statistics to identify OTUs were run using the PHYMYCO-DB database (*Mahé et al., 2012*). A contingency table was produced to perform all the diversity and statistical analyses. Even though the difference in the number of sequences among samples was below 10%, the dataset was rarefied to the same number of sequences using the module VEGAN (*Oksanen et al., 2015*) in R (*R Core Team, 2014*) before statistical analysis. All sequences were uploaded in the European Nucleotide Archive with the accession number PRJEB22764.

## The effect of *Arthrinium phaeospermum* on rice growth

In order to assess the effect of one of the fungi associated with yield under drought in the present study, the endophytic fungus *Arthrinium phaeospermum* was used in a pot experiment to study its effect on rice performance. As the original *A. phaeospermum* strain from the field was not isolated at the time that the experiment was done, eight strains of the species that were available from the CBS-KNAW Fungal Biodiversity Centre (Utrecht, The Netherlands) were tested (Table S4). In total we sowed 144 plants (eight replicates per treatment and fungal strain). As host, the cultivar IR36 (indica rice) was selected, because this cultivar had a higher *A. phaeospermum* presence in our field experiment. The seed husk was removed and seeds were sterilized with 2% sodium hypochlorite (v/v) and rinsed several times in sterile distilled water. Seeds were directly sown in small 0.3 liter (L) pots filled with sterilized sand. Plants were watered regularly with modified half-Hoagland nutrient solution and grown during seven days in a climate cell at 28 °C/25 °C and a 12 h photoperiod at 75% relative humidity and a light intensity of 570 $\mu$moles m$^{-2}$ s$^{-1}$. The fungal cultures were grown in Potato Dextrose Agar (PDA) with rifampicin (50 $\mu$g/ml). After the fifth day, the upper part of the soil from the pot close to the plant root was inoculated with a 10 mL diameter agar disc with mycelium, then covered with a bit of soil and grown for another two days when the drought treatment was started, which consisted of water withholding for six days. To avoid that the plants died, they received a fixed amount of water every day (until 50–55% of field capacity) to keep the stress high but not to lose all plant available soil water. After the drought period, all plants were collected and fresh and dry weights were quantified. The hyphae colonization was checked under the microscope in some of the samples for a qualitative purpose.

## Statistical analysis

All the statistical analyses were performed using R (R core team, 2013). From the contingency matrix, OTU richness (number of species), abundance (number of individual OTUs), evenness and diversity index (Shannon H′index) estimators were calculated using the VEGAN (*Oksanen et al., 2015*) and BIODIVERSITYR (*Kindt & Coe, 2005*) packages. Statistical differences in these measures were analyzed using ANOVA, with the treatments (control and drought) as factors using the CAR package (*Fox & Weisberg, 2011*). To test for a field position effect on the microbial community results, a Mantel Test and correlogram analysis were performed using the VEGAN package. Each root sample was assigned a field position value (based on two coordinates) and the geographical Euclidean distances were calculated. These distances were subsequently compared with the ecological distances (Bray–Curtis method) calculated for the fungal community to analyze if there is a correlation between the field position and the fungal community distance.

Fungal community differences between the different treatments were studied using non-metric multidimensional scaling (NMDS) analysis, after removing rare OTUs (OTUs with < 10 sequences) using the Bray–Curtis statistic to quantify the compositional dissimilarity (*Kulczynski, 1928*). To test whether significant differences exist between fungal communities from control and drought treatments a permutational multivariate analysis of variance

(PERMANOVA) was run with the "adonis" function using the NMDS factor scores (VEGAN Package).

To study the correlation between plant performance and the associated fungal community, a Variation Partitioning analysis (VPA) was performed in VEGAN using the "varpart" function. The VPA model allows to include many factors as variables to study if they can explain the fungal community composition. In the model the OTU relative abundance data (without the rare OTUs) were included as response variable and 'yield' (described by the grain in grams per square meter) and the rice 'host' (described by the Kinship values from the rice genomic map (*McCouch et al., 2016*)) as explanatory variables. As a way to calculate the relative response between treatments, the 'yield robustness' was calculated by the phenotypic plasticity index (PI) (*Valladares, Sanchez-Gomez & Zavala, 2006*) defined as (yield$_{control}$ − yield$_{drought}$)/yield$_{control}$ (calculated for each cultivar). This index was included as an explanatory variable together with the 'host' factor in a new VPA model to study how yield robustness under drought is correlated with the community. We also ran a Spearman correlation analysis with the *rcorr* function in the HMISC package, between the independent OTUs and yield under control and drought treatments; the OTUs positively correlated with plant yield with a $P < 0.004$ were selected for further phylogenetic analyzes, as results with P-values below this threshold were not significant (the *P*-value cutoff was a result of the correction for multiple testing).

When exploring changes in fungal communities from OTU patterns of plants fungal microbiota exposed to drought conditions, the use qualitative and discrete quantification methods are useful to limit the possibility that changes in community composition (OTUs) be blurred by differences in OTU abundance (*Lozupone & Knight, 2008*; *Amend, Seifert & Bruns, 2010*; *Magurran, 2013*). Hence, we also estimated the OTU occurrence (presence/absence) in the different treatments for the OTUs positively correlated with yield.

To study if yield is linked to phylogenetic relatedness of the root-fungal microbiota, the phylogenetic signal was calculated using the Blomberg's K statistic, which compares the observed signal in a trait to the signal under a Brownian motion model of trait evolution on a phylogeny (*Blomberg, Garland & Ives, 2003*) with the PICANTE package (*Kembel et al., 2010*). The OTU relative abundance matrix was used as a trait, where the mean and standard error was calculated for each OTU. The original Ascomycota tree generated by Maximum Likelihood Estimation was pruned by the yield correlated OTUs. The pruned tree together with the OTUs abundance data was used to calculate the phylogenetic signal.

Pruned trees (i.e., where OTUs with less than 10 sequences had been removed) were separately calculated for the main phyla, Ascomycota and Basidiomycota. Sequences were aligned using MAFFT v.7.123b (*Katoh & Standley, 2013*) and then trimmed with Gblocks v.0.91b (*Castresana, 2000*). Phylogenetic trees were generated by Maximum Likelihood (ML) using RAxML v.8.00 (*Stamatakis, 2014*), with the General Time Reversible (GTR) model of nucleotide substitution under the Gamma model of rate heterogeneity and 1,000 bootstrap replicates. For a subset of OTUs correlated with yield, a Neighbor Joining (NJ) tree was generated from a pairwise distances matrix of sequences using the SEQINR (*Charif & Lobry, 2007*) and APE (*Paradis, Claude & Strimmer, 2004*) R packages. All trees were edited using iTOL (http://itol.embl.de, *Letunic & Bork, 2011*).

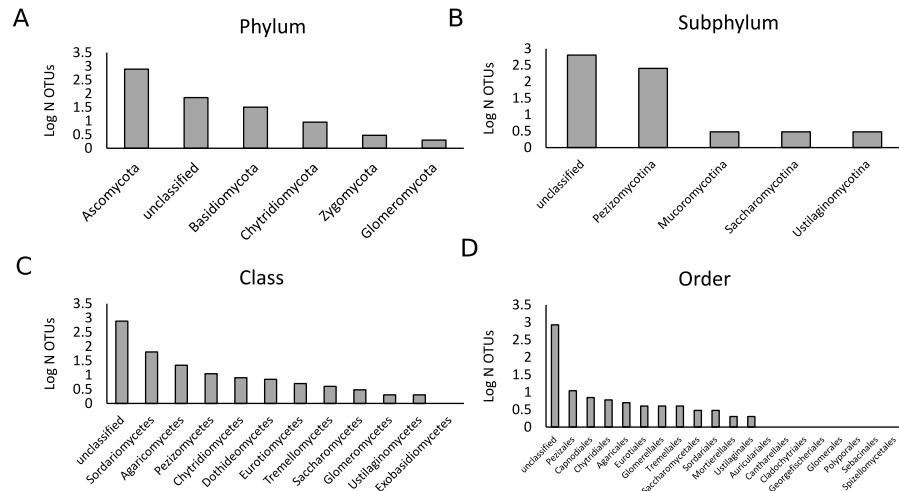

**Figure 1 Description of the fungal community in the present study.** A total of 447,757 sequences were analyzed that belonged to 902 OTUs. Bars represent the logarithmic value of the number of OTUs per taxonomic group. The OTU richness per phylum (A), subphylum (B), class (C) and order (D) are shown.

To analyze the effect of *Arthrinium phaeospermum* on plant productivity in our pot experiment, a linear model analysis was performed using the STATS package. The response (plant biomass, water content, root to shoot ratio) and the predictors (treatment 'fungus' and treatment 'drought') were included in a fitted linear model that was then used to run an ANOVA analysis.

All data and code for the analyses are available as supplementary material.

## RESULTS

### Root—fungal microbiota in rice

As the samples were selected from a large field experiment, we performed a Mantel Test to check for the presence of field position effects. This analysis showed that there was no strong effect of field position on the fungal community composition for both treatments (Fig. S1). We analyzed a total of 444,757 fungal sequences of 560 bp forming 902 different OTUs (Fig. 1). The sequencing depth was sufficient to describe the root fungal microbiota (Fig. S2). The 18S rRNA marker has been shown to provide adequate species-level resolution for the identification of many fungal groups, with the exception of the Ascomycota (*Vandenkoornhuyse et al., 2002*). Despite the use of the fungal 18S rRNA gene database PHYMYCO-DB (*Mahé et al., 2012*) and its better resolution compared to more generalists databases to identify fungal sequences, most of the OTUs did not match to curated sequences of known close relatives (i.e., they are unknown at the species level or higher taxonomic ranking). Among the 902 OTUs detected, only two belonged to the Glomeromycota (i.e., AM fungi). The biggest OTU richness by far was observed for the Ascomycota phylum (784 OTUs), followed by the Basidiomycota (32 OTUs) (Fig. S3). The remaining OTUs belonged to the Chytridiomycota (nine OTUs), Zygomycota (3 OTUs) and an unclassified phylum (72 OTUs). After filtering out the rare OTUs (here defined as

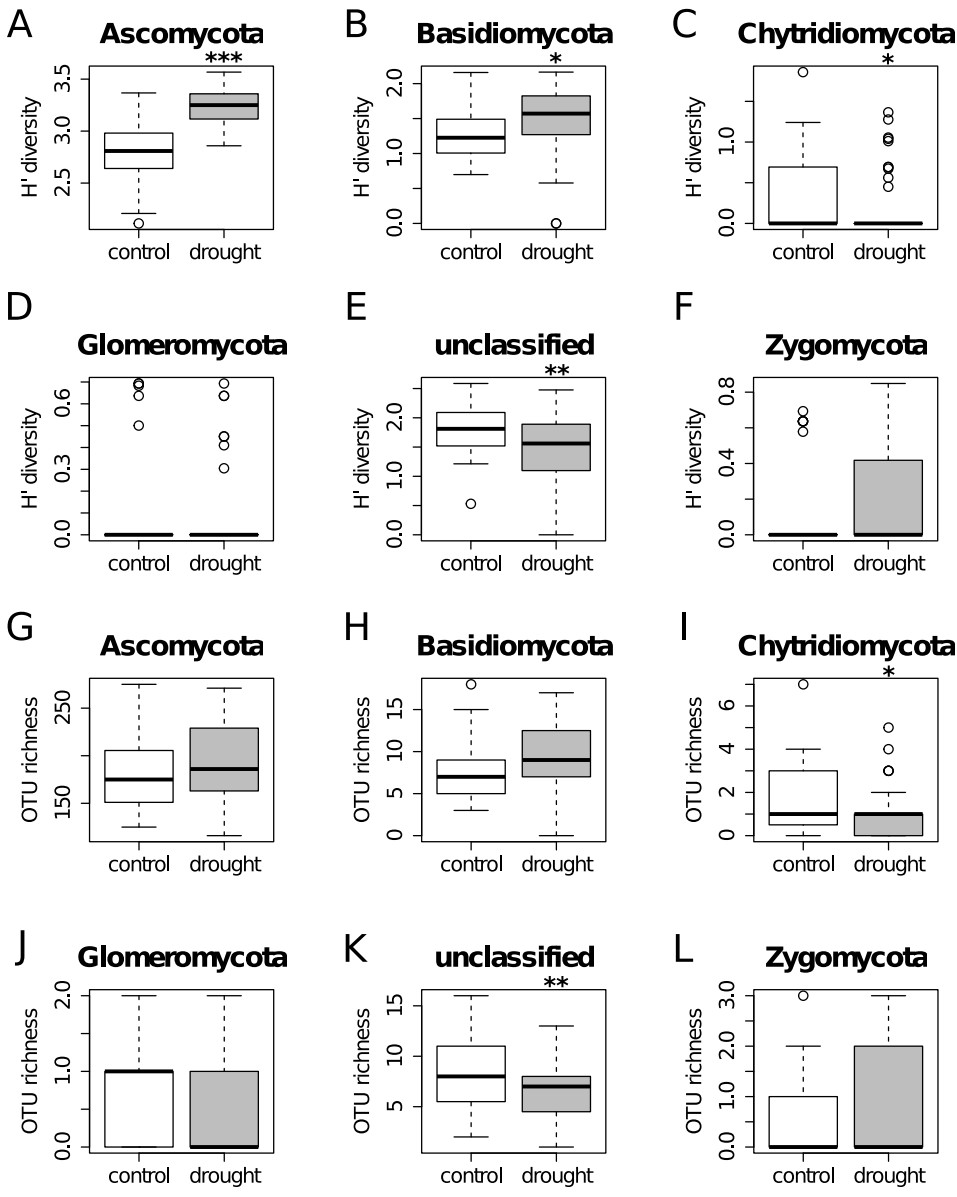

**Figure 2** Diversity Shannon index (A–F) and richness (G–L) for the different phyla, under control (white) and drought (grey) conditions (i.e., α-diversity). Results show that OTU richness do not differ much between treatments for all the taxa. On the other hand, diversity is higher under drought for Ascomycota and Basidiomycota, while the unclassified group shows the opposite trend.

OTUs with less than 10 sequences in all analyzed samples), the fungal γ-diversity measure, S, was 862 and the Shannon diversity index, H′, was 3.5. The γ-diversity in the different treatments was similar, and the majority of OTUs are present under both control and drought (Fig. S4).

The OTU richness and diversity per taxonomic group differ between the control and drought treatment (Fig. 2). The diversity and OTU richness for the main groups

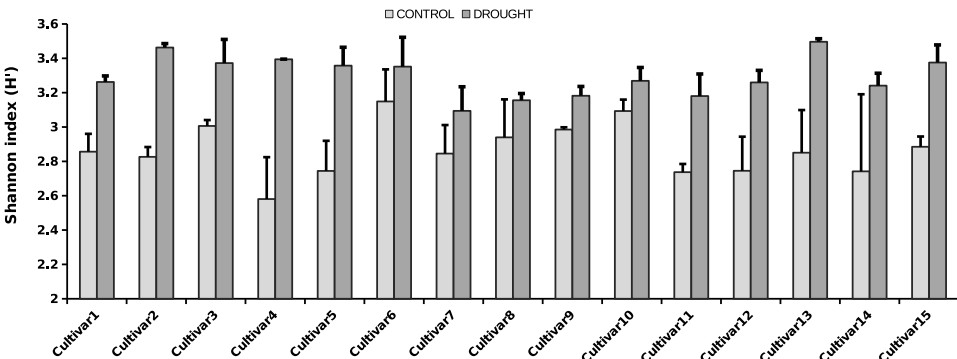

**Figure 3   Shannon diversity index for the rice cultivars analyzed, under control (light grey bars) and drought (dark grey bars) (i.e., α-diversity).** Error bars represent SE. The fungal microbiota Shannon index strongly differs between the treatments (i.e., two-way ANOVA analysis, $P < 0.001$).

(Ascomycota and Basidiomycota) were higher under drought, whereas the unclassified phylum showed the opposite pattern. Using α-diversity, there were small differences in fungal microbiota OTU richness under control and drought, both with non-normalized as well as with normalized data: $S_{control} = 124$, $S_{drought} = 132$. An uneven distribution of OTUs in the rice fungal microbiota community structure was observed ($J_{eveness}$ index ~0.5). This observation matches with the Shannon diversity index (H′), which was higher under drought for all the rice cultivars (Fig. 3), due to an increased OTU richness and the presence of less dominant species. This was confirmed by two-way ANOVA analysis ($P = 9.7 \times 10^{-13}$; $F = 71.08$; $Df = 1$). Interestingly, the magnitude of the change in diversity between control and drought was rice cultivar-dependent (Fig. 3) suggesting an effect of the host-plant on fungal biodiversity. Community compositions differed significantly between treatments (Fig. 4). A phylogenetic analysis of all frequent OTUs (without the rare OTUs) was performed for the main phyla: Ascomycota and Basidiomycota (Fig. S3). OTUs within the Sordariomycetes (Pezizomycotina) and an unclassified group (closely related to Sordariomycetes) dominated (Fig. S3).

To test the statistical significance of host genotype and treatment visualized with the NMDS analysis, a PERMANOVA analysis was performed on the NMDS scores. The NMDS analysis was based on the dissimilarity matrix (Bray–Curtis), but using the rank orders rather than absolute distances for the PERMANOVA gave us less biases linked to data transformation. With both data (Bray–Curtis dissimilarity matrix and NMDS scores) the results were the same. The analysis supports that there is a strong effect of the treatment (control *vs*. drought) ($R^2 = 0.37$; $P = 0.001$) (Fig. 4). In conclusion, the data show that rice genotype and drought have a qualitative and quantitative impact on the fungal community associated with the roots.

## Host and treatment effect on root fungal microbiota

To further underpin the effect of drought on the fungal community composition we used Variation Partitioning analysis (VPA). This analysis compares the root associated microbial community with factors or a group of factors and tests if any of them is correlated
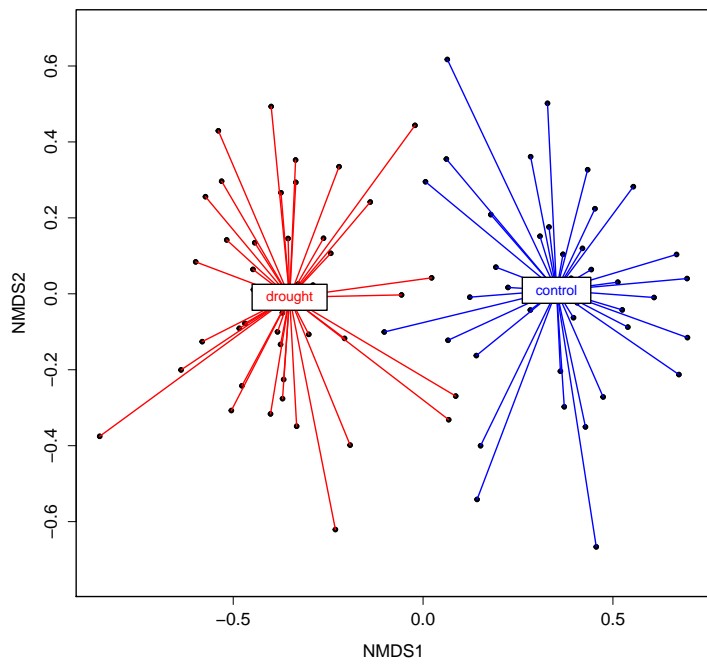

**Figure 4** **NMDS representing rice root fungal community structure.** A Bray–Curtis dissimilarity distance (i.e., β-diversity) and a Kulczynski ordination method were used. The statistical analysis (PERMANOVA) showed that the treatments significantly differed in the fungal microbiota composition ($R2 = 0.37, P = 0.001$).

with the microbial community structure. In a first VPA model the factors 'treatment' (control/drought), 'host' (genotype Kinship values) and 'yield' were included. Both the 'treatment' effect and the combination 'yield' and 'treatment' significantly explained the variation in fungal community composition (i.e., response matrix) ($P = 0.001$; coefficient of determination, $R^2$, of 0.22 and 0.38, respectively) (Fig. S5A). We observed a similar result using the PERMANOVA analysis. The 'host' effect was very small in the VPA analysis ($R^2 = 0.01$), also confirming the PERMANOVA analysis. In a second VPA analysis, we included 'yield robustness' along with the factor 'host' and the abundance of the OTUs for the different treatments (control and drought) and demonstrated a significant 'host' effect on the fungal community under drought ($P = 0.002$; coefficient of determination $R^2 = 0.13$) while 'yield robustness' gave no significant effect (Fig. S5B). Also 'yield robustness' and OTU abundance under control showed a significant 5% of explanation by the 'host' ($P = 0.05$) but not by 'yield robustness'. Thus, fungal community under a stress environment seems to be more relevant for plant yield robustness than when normal conditions.

## Effect of fungal endophytes on rice fitness

To address the link between the fungal community and plant fitness under drought, each independent OTU was correlated with seed yield (control and drought separately) as a proxy for drought tolerance. We found 37 OTUs that were positively correlated with

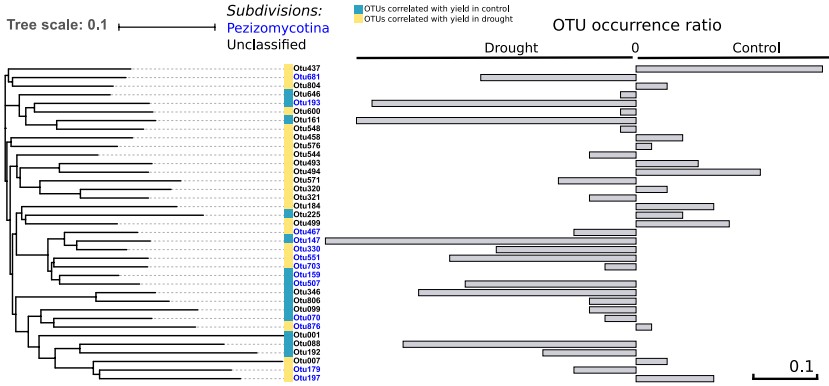

**Figure 5 Phylogenetic tree.** It represents the 37 OTUs positively correlated with yield under control and drought conditions. The represented OTUs present a correlation value of $R > 0.30$ with a $P < 0.004$. The grey bars provide the OTU occurrence (presence or absence) ratio between treatments: OTU occurrence control—OTU occurrence drought. The occurrence of only two of the 37 OTUs remained unchanged between treatments while 22 of the 37 OTUs increased under drought. There is a strong phylogenetic signal between all yield correlated OTUs ($K = 6.6$; $P = 0.01$), indicating that yield correlated OTUs are related.

yield in both treatments ($R > 0.30$; $P < 0.004$), of which 13 were occurring more under control and 22 more under drought conditions –which therefore are candidates to have a positive effect on drought tolerance—while of two the presence did not change between the treatments (Fig. 5). Thirteen out of the 37 OTUs were assigned to the Pezizomycotina while the other 24 OTUs could not be classified, although they are closely related to the Pezizomycotina sub-phylum.

Comparing the phylogenetic signal for yield robustness for each OTU in comparison with OTU abundance showed that there was phylogenetic conservation for yield ($K = 6.6$, $P = 0.01$). This means that phylogenetically related OTUs are more associated with similar yields than random OTUs. This relatedness is solely due to the data under drought ($K = 8.7$; $P = 0.03$).

One of the OTUs identified at the species level, *Arthrinium phaeospermum*, was among the ones contributing significantly to plant yield ($R = 0.08$; $P = 0.01$) and yield robustness ($R = 0.15$; $P = 0.01$) in the VPA analysis. We found other Sordariomycetes (e.g., *Chaetomium sp.*), Saccharomycetes and Dothideomycetes that also were associated with increased plant yield under drought. Interestingly, *Arthrinium phaeospermum*, belongs to the Pezizomycotina subphylum, which is a group that includes the majority of beneficial fungal endophytes, and the species has been described to promote plant growth (*Khan et al., 2008*). Therefore, we decided to study it in further detail and used a pot experiment to study its effect on rice. Since we did not have access to sufficient field-collected material for isolation of the corresponding field strain, we ordered six different *A. phaeospermum* strains from CBS and tested their effect on rice growth under control and drought conditions. The *A. phaeospermum* strains tested did not have a significant positive effect on the plant shoot biomass under control nor drought conditions (Table S1). We did see an interaction between the factors 'fungus' and 'drought' for the majority of variables measured (Table S1).

Indeed, the majority of the fungal strains reduced root biomass under drought (Fig. S6) and affected the root to shoot ratio significantly in the case of strains 2, 4, 7 and 8 (Table S2).

## DISCUSSION

### Endospheric fungal microbiota detection

There is an increased understanding of the complexity of the root fungal microbiota which is not solely limited to Glomeromycota forming AM association, but also includes other fungi belonging to the Zygomycota, Ascomycota and Basiodiomycota (e.g., *Vandenkoornhuyse et al., 2002*; *Lê Van et al., 2017*). In the present study, we report for the first time the analysis of the whole fungal microbiome associated with the roots of rice in the field. The largest group of OTUs we detected was the Ascomycota phylum (784 OTUs), followed by the Basidiomycota (32 OTUs) (Fig. S4). The Ascomycota and Basidiomycota are also dominant in the roots of other plant species such as maize (*Kuramae et al., 2013*), wheat (*Vujanovic, Mavragani & Hamel, 2012*), poplar (*Shakya et al., 2013*) and *Agrostis stolonifera* (*Lê Van et al., 2017*), and they are known to include "dark septate endophytes" (DSEs), which are facultative plant symbionts (*Rodriguez et al., 2009*).

In this study, the diversity values (H′ = 3, 5; S = 862) are of the same order of magnitude as in other crops. We found a lower H′ and different community structure than in chickpea for which a H′of about 4.7 and S of about 800 have been reported (*Bazghaleh et al., 2015*) but a higher H′and S than in arctic plants for which an H′ of 2.8 and S of 60 have been reported (*Zhang & Yao, 2015*). For other monocots such as wheat: H′∼1.8; S∼18, and maize: H′∼0.9; S∼9 (*Bokati, Herrera & Poudel, 2016*) the values are also quite a bit lower than our values, although for the latter the fungal community analysis was done in a very different way. Thus, the rice genotypes used in the present study appeared to recruit a rather high number of fungal species. It is possible that host defense was lowered due to stress and/or plants signaled for help, which resulted in additional fungal species to colonize the roots. The high OTU richness found in the rice root fungal endosphere when compared with other studies, could also be an effect of the primer choice or could be related to the fact that rice is growing in a very different and specific environment in comparison to the other plant species (i.e., in the tropics in a water saturated agroecosystem).

### Drought affects the endophytic fungal microbiota

It has been reported that the soil fungal community composition changes under drought resulting in a decreased α-diversity (*Hawkes et al., 2011*; *Cregger et al., 2012*; *Sharma & Gobi, 2016*; *Zhang et al., 2016*). As far as we know, the consequence of drought on the root associated fungal microbiota has not been investigated before under field conditions. In the present study we clearly demonstrate that the rice endospheric fungal microbiota composition changes under drought stress (Fig. 4) and results in an increased richness of fungal OTUs within rice-roots for all the 15 rice cultivars tested (Fig. 3). Increased fungal richness could be interpreted as an active recruitment of additional fungi by the rice root to face the environmental stress although we cannot exclude that this is the result of the reverse process: fungi actively colonize the root compartment to escape from the drought effect. Nevertheless, a higher fungal diversity could represent a better pool for subordinate species

(less abundant ones), which may have a large influence on certain ecosystems and can potentially improve plant productivity under drought conditions (*Mariotte et al., 2015*). The increase in fungal species richness may result in the enrichment in additional functions enabling to mitigate the consequences of drought on the host plant. Also, other studies suggest that fungi have an important effect on plant fitness under drought conditions (*Lau & Lennon, 2012*; *Kaisermann et al., 2015*; *Classen et al., 2015*). In sorghum it has been shown that when water levels are extreme (drought or flooding), roots are colonized by fewer AM fungal species, however at the same time the abundance of these species increases probably because they are more adapted to the new conditions. In those experiments, plant biomass was not affected by the water regime, but phosphate uptake was increased as a result of a change in the root colonization of plants under non-flooded conditions (*Deepika & Kothamasi, 2015*).

Glomeromycota species richness and abundance increased under drought within a diverse panel of plants including wild and cultivated species (*Tchabi et al., 2008*). Strikingly, in the present study, we only observed two OTUs representing Glomeromycota within the fungal microbial community and they were not affected by drought. Although we know that the fungal microbiota is not only composed of Glomeromycota (e.g., *Vandenkoornhuyse et al., 2002*), in our experiment rice is unexpectedly poor in AM fungal colonizers in comparison to other Poaceae. For example, in a study on *Agrostis stolonifera* and using the same methodological approach as in the present study, the Glomeramycota represented 10% of the root fungal microbiota (*Lê Van et al., 2017*). As already commented in the Introduction, monocropping and conventional paddy cultivation have been shown to reduce the AMF diversity and colonization in rice, which likely explains the low Glomeramycota representation in the present study.

The majority of the OTUs that increased in frequency under drought in our study belong to the Pezizomycotina subphylum, the most abundant subphylum in the Class II fungal endophytes (*Rodriguez et al., 2009*). They are well-known for their role in plant performance, boosting plant growth and buffering the effect of environmental stresses and protecting their host against pathogens (*Maciá-Vicente et al., 2009*; *Jogawat et al., 2013*; *Azad & Kaminskyj, 2015*). If looking at other individual OTUs there are changes in their abundance between treatments and/or rice cultivars; however, those changes are not following a pattern as a taxonomic group or the description we get at species level is not enough to make further conclusions.

## Host genotype affects the fungal microbiome response to drought

Using VPA we showed that the host genotype affects the structure of the root associated fungal community, also in response to drought ('host' effect: $R^2 = 0.13$; $P = 0.01$) (Fig. S5). Previous studies using *Arabidopsis thaliana* and barley also show a host-genotype effect on the root associated microbiome (*Lundberg et al., 2012*; *Bulgarelli et al., 2015*), However, in maize and *Microthlaspi spp.* the rhizosphere community composition did not depend much on the host genotype, but was largely determined by the geographical distribution where these cultivars are coming from (*Peiffer et al., 2013*; *Glynou et al., 2016*). Using a GWAS approach for the phyllosphere microbiome composition of *Arabidopsis thaliana*, it

was shown that the fungal and bacterial community on leaves is determined at least in part by plant genomic loci, in this case by loci responsible for defense and cell wall integrity (*Horton et al., 2014*). Recently, a new study has shown that drought induces changes in the root bacterial and fungal endophytic community in four rice cultivars under greenhouse conditions (*Santos-Medellín et al., 2017*), supporting what we observe in our study in the field.

The results of the present study clearly show that changes occur within the fungal microbiota community composition when plants experience an environmental constraint (Fig. 4). The increased root fungal endophytic diversity could be the result of migration of soil fungi to the roots to survive the drought conditions. However, the significant genotype effect on the fungal community structure under drought (Fig. S5), strongly suggests that active recruitment by the plant host of fungal species (also) occurs. Potentially, this enrichment of plant-microbiota can buffer the effects of the drought stress (*Vandenkoornhuyse et al., 2015*). A host-plant preference has also been shown in studies analyzing AM fungal communities (*Martínez-García & Pugnaire, 2011*; *Torrecillas, Alguacil & Roldán, 2012*) even among co-occurring plant species within the Poaceae (*Vandenkoornhuyse et al., 2003*). This observation was later explained by the ability of plants to filter the colonizer by a carbon embargo toward less beneficial AM fungi (*Kiers et al., 2011*; *Duhamel & Vandenkoornhuyse, 2013*). We are currently further exploring the role of the rice plant-host in the recruitment of root-associated fungal microbiota using plant genetics approaches.

## Root fungal microbiota and rice grain yield

OTUs that are closely related to each other showed similar correlation values with rice grain yield as there is a strong phylogenetic signal between all yield correlated OTUs ($K = 6.6$; $P = 0.01$). Intriguingly, these OTUs are more abundant under drought (Fig. 5), suggesting that they may play a role in the tolerance of rice to drought. In an earlier study, inoculation of rice with fungal Type II endophytes such as *Fusarium culmorum* and *Curvularia protuberata* resulted in a higher growth rate and yield and a reduced water consumption. Moreover, the rice plants grown under drought stress were more intensively colonized by these fungi in comparison to control plants (*Redman et al., 2011*). In the present study we identified 37 different OTUs that belong to the Pezizomycotina which all positively correlated with yield in plants that were exposed to drought (Fig. 5). This might be due to one particular fungal OTU or alternatively might be the consequence of a complex synergistic effect of different OTUs.

Among these fungi there was *Arthrinium phaeospermum*. *Arthrinium* species are often associated with plants from the Poaceae family, suggesting a certain level of host specificity (*Yuan et al., 2011*). To confirm the role of *A. phaeospermum*, different strains of this species were used in a pot experiment. Under control conditions no significant effect of the inoculation was observed on plant shoot biomass, while root biomass was decreased by some of the strains under drought (Table S2). Root biomass investment (root to shoot ratio) under drought was lower for plants inoculated with some of the strains (Table S2; $P < 0.05$). These results are counter-intuitive because in the community analysis, *A. phaeospermum*

was correlated with yield, especially under drought as shown by the VPA analysis. The most likely explanation for this is that we did not use the *A. phaeospermum* strain that caused the effect in the field because we used publicly available strains. Also, in the pot experiment biomass was analyzed instead of yield. Another possible explanation is that the OTU we described as *A. phaeospermum* is actually a different, though closely related, species. To further examine this discrepancy, it will be necessary to isolate the corresponding strain from the field and/or plant material. Another possible explanation is that the yield effect is not directly due to *A. phaeospermum* but to other microorganism(s) that were not analysed in our study (e.g., bacteria) that are correlated with the presence of *A. phaeospermum*. Drought tolerance may be the result of a synergistic/antagonistic effect between *A. phaeospermum* and these other microorganisms (*Larimer, Bever & Clay, 2010*; *Aguilar-Trigueros & Rillig, 2016*), while we studied the effect of a single fungal isolate. Likewise, a perturbation of the root microbial community induced by the inoculation may have blurred any positive effects.

A higher root:shoot ratio and a longer root length are often characteristics for rice cultivars that are more drought tolerant, as they are good indicators for a higher water uptake capacity (*Comas et al., 2013*; *Paez-Garcia et al., 2015*). We did not record the root length in the pot experiment, so it could be that some of the fungal strains may have had an impact on root length rather than on root biomass. Furthermore, the effect of drought on the root to shoot ratio depends on the plant growth stage, which is most evident in older plants (*Silva, Kane & Beeson, 2012*). Therefore, in the relatively young plants that were used in the present study we may have missed the effect that the fungi may have on root architectural changes in older plants. These possibilities should be considered for future studies with the same research questions.

## CONCLUSIONS

Our study illustrates that the root associated fungal community in rice changes under drought, resulting in a higher species diversity in the rice-root endosphere. It also shows the presence of specific OTUs (belonging to the Pezizomycotina) is correlated with yield, and the relative abundance of these OTUs increases under drought. Finally, we also show that, under drought, the rice genotype has a significant effect on the fungal community composition.

Roots are interesting to search for beneficial-plant growth promoting fungi (*Fonseca-García et al., 2016*; *Angel et al., 2016*). With sufficient knowledge, we can potentially compose 'functional OTU clusters', specifically tailored for a crop plant species, that we know may have a positive impact on plant performance. This microbial consortium could then be applied in the field to boost plant productivity under periods of stress. However, only a maximum of 1.0% of soil microorganisms can be cultured under standard conditions. Thus, studying the roles of microbiota in biological and ecological soil processes remains a challenge (*Rehman, Akhtar & Abdullah, 2016*), especially for possible application in agriculture. Nonetheless, metagenomics and metabarcoding studies can yield valuable information that could help us to exploit microbial communities and further investigate

how microbial 'clusters' are working together to improve plant fitness under stressful environments.

## ACKNOWLEDGEMENTS

We thank support staff of IRRI for their help with sample collection and processing and the Human and Environmental Genomics platform (https://geh.univ-rennes1.fr/) and S. Michon-Coudouel for technical support in the library preparation and sequencing, and J.G. Maciá-Vicente for providing R scripts for some of the statistical analyses and his support with some of the phylogenetic analyses.

### Funding

This work was supported by a private donor (who is known to us but has requested anonymity) who requested that their funds be applied to this project via the Wageningen University Fund. To the best of our knowledge this donor has no conflicts of interest related to this project. The work was also funded by a grant 'défis émergents' from the University of Rennes 1. The funders had no role in study design, data collection and analysis, decision to publish, or preparation of the manuscript.

### Grant Disclosures

The following grant information was disclosed by the authors:
Wageningen University Fund.
défis émergents.

### Competing Interests

The authors declare there are no competing interests.

### Author Contributions

- Beatriz Andreo-Jimenez conceived and designed the experiments, performed the experiments, analyzed the data, prepared figures and/or tables, authored or reviewed drafts of the paper, approved the final draft.
- Philippe Vandenkoornhuyse contributed reagents/materials/analysis tools, authored or reviewed drafts of the paper, approved the final draft.
- Amandine Lê Van analyzed the data.
- Arvid Heutinck performed the experiments, analyzed the data.
- Marie Duhamel contributed reagents/materials/analysis tools.
- Niteen Kadam performed the experiments.
- Krishna Jagadish conceived and designed the experiments.
- Carolien Ruyter-Spira conceived and designed the experiments, authored or reviewed drafts of the paper.
- Harro Bouwmeester conceived and designed the experiments, authored or reviewed drafts of the paper, approved the final draft.

## Data Availability

All sequences generated are available in the European Nucleotide Archive: PRJEB22764.
https://www.ebi.ac.uk/ena/data/view/PRJEB22764.

## Supplemental Information

Supplemental information for this article can be found online at http://dx.doi.org/10.7717/peerj.7463#supplemental-information.

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
