# Peer review of "Plant host and drought shape the root associated fungal microbiota in rice"

_PeerJ, doi:10.7717/peerj.7463_

## Round 0.1 · original submission · Minor Revisions

The reviewers commented very positively on your manuscript and its importance to understanding the impact of the root microbiome on plant health under environmental stress. However, the reviewers and myself noted a number of missing information and/or analyses in the study rendering it difficult to justify all of the conclusions. Thus, I encourage you to respond positively to these comments in revising the manuscript.

Reviewer 1 ·

Basic reporting

Clear language was used throughout unless otherwise noted.

Sufficient background, literature, and context provided.

Good article structure.

Specific Comments:

This is not the first study to analyze the fungal microbiome in rice when exposed to drought, as the authors state in lines 359-360 and 383 - 384. Santos et al, 2017 (which the authors have cited) conducted a greenhouse experiment analyzing the bacterial and fungal microbiome. I certainly understand the authors missing that the fungal microbiome was analyzed in the Santos study. The analysis is almost hidden as the last supplemental figure in the paper and it was not heavily discussed. I only know this because I was a co-author on the paper. There are nice similarities between the two analyses, and this certainly does not take away from the strength of the manuscript here. Andreo-Jimenez et al. conducted a field experiment (compared to a greenhouse experiment) and used 18S sequencing, which in my opinion is a better barcode than ITS. The authors include 17 genotypes (compared to 4). The authors only need to adjust the language here to reflect how their study substantially expands upon our knowledge of the rice fungal microbiome under drought and well watered conditions (which it definitely does).

I am very confused by the paragraph on lines 330-334. It could use rewriting.

Line 386: I think the word “enrichment” is misleading here. The authors intend to mean an increase in alpha diversity, but enrichment could mean other processes such as overall abundance of fungi are increasing under drought or specific OTUs are increasing under drought. The authors have not shown whether either of these scenarios are happening, therefore they should be more precise when discussing this as an increase in alpha diversity.

Experimental design

This is original primary research that falls within the Aims and Scope of PeerJ

The research question is well defined and definitely meaningful.

Rigorous investigation was performed.

Specific Comments:

An interesting aspect to this manuscript that remains under analyzed is the effect of genotype. The authors report a significant effect in the variance partitioning, but it would be nice to visualize the difference between genotypes in drought and control conditions separately. For example, a PCoA, NMDS, or CAP ordination for each treatment would be nice to see. Do the similarities between the genotypes change depending on the treatment? Is the genotype effect more evident under drought or control conditions, i.e. is there a significant interaction term between watering treatment and genotype in the PERMANOVA? Is there one genotype which is very divergent (in terms of fungal microbiome structure) compared to the others? I think these are all interesting questions that geneticists and breeders will ask when they see the experimental design.

I think it would also be possible to factorize the rice genotypes in the authors analysis, rather than using kinship values. For example, the genotype variable could be coded [G1, G2, G3...Gn] rather than having numerical values. It is unclear in the literature (at least from what I have read) the extent to which bacterial and fungal microbiomes correlate with relatedness - rather than simply varying between genotypes. For instance two closely related genotypes may differ substantially, while two distantly related genotypes may host microbiomes that are relatively similar. I think this should be an optional analysis for the authors to conduct - but still it interests me greatly.

Validity of the findings

In this manuscript, Andreo-Jimenez et al. survey the fungal portion of the root-associated microbiome of field grown rice varieties in control and drought conditions. The authors find that drought stress impacts the composition of the root fungal microbiome, with drought stressed roots hosting a fungal microbiome with greater diversity than control plants. They also find that genotype significantly affects the composition of the fungal microbiome. Because the authors collected yield data for the selected rice varieties, they were also able to correlate the abundance of specific fungal taxa with grain yield. After finding interesting correlations, the authors used isolates of Arthrinium phaeospermum to examine how root and shoot phenotypes would be impacted by the addition of putatively beneficial fungi. The authors found the A. phaeospermum isolates affected root growth negatively. I think these are very interesting results and are relevant to plant breeders and microbial ecologists alike.

Additional comments

Great job. I really like the structure of this study from going to the field and collecting microbiome and plant yield data and then trying to take what was learned from the study to follow up in the greenhouse. I think the field could use the structure of this study as an example. It is a shame that the fungal isolates did not yield conclusive results - but I like how the authors discussed the various possibilities around that and how to move forward with it.

Reviewer 2 ·

Basic reporting

ARTICLE STRUCTURE.
Overall, the paper is clearly written and well organized. I only have a few comments that would hopefully improve the interpretability of some of the figures:

Fig.1.
Even though the y-axes in all panels are labeled the same, it seems the actual values displayed are only log-transformed in some of them. In particular, the values on the y-axis in the Phylum panel clearly represent the actual number of OTUs displayed on a log10 axis (the sum is ~900, which matches the reported number of OTUs found in the study), while the numbers displayed on the y-axis for the rest of the taxonomic ranks seem to be log-transformed values. For consistency, I would suggest to use a uniform format to avoid any confusion.

Fig. 2.
Indicate which groups showed statistically significant differences between drought and control communities.

Fig.5.
Label the occurrence ratio axis.

DATA AVAILABILITY.
The raw data is available, although I would also encourage the authors to share their scripts to facilitate the reproducibility of the analyses described in the paper.

Experimental design

RESEARCH QUESTION.
The authors tackle a relevant question in the field of plant microbe interactions: how do environmental stresses affect the community structure of the plant-associated microbiota and how does the host genetic makeup shape this response. By including 15 rice varieties, the authors generated a useful dataset to delve into this problem. While they found some interesting patterns in the alpha and beta diversities of these communities, I wished they would've expanded on some other aspects. For example, even though the authors found that drought significantly shifted the overall composition of the fungal microbiota of rice roots, they didn't go into a more detailed analysis of the changes in relative abundances of individual OTUs responding to this environmental stress. A small discussion about drought-mediated changes in frequency (Fig. 5), yet this analysis is restricted to the OTUs positively correlated to yield. In my opinion, including a section describing the differential OTU abundance patterns between drought-stress and well-watered communities could further improve the breadth of this study: how many OTUs are driving the changes in microbiome structure? are there any taxonomic patterns behind drought-enriched or drought-depleted OTUs? Are the changes in relative abundance affected by the host genotype? Similar analyses have been recently applied to drought-stressed root bacterial communities on rice (Santos-Medellin et al. mBio 2017) and other grasses (Naylor et al. ISME 2017).


METHODS.

- Many of the results discussed in this manuscript are based on the patterns of presence/absence of OTUs between the various experimental factors (e.g., richness, OTU overlap between treatments, and occurrence ratios). Yet, there is no mention as to how the authors dealt with the potential differences in sequencing depth across samples. While they filtered out low-abundance OTUs (those with less than 10 sequences in the whole data set), this approach doesn’t account for uneven total sequence counts between libraries. If there are big differences in sequencing depth between samples, this could complicate the interpretation of diversity patterns as the absence of OTUs could be a consequence of relatively shallow sequencing in some libraries. An option to deal with this issue is to rarefy the libraries to a uniform depth (see Weiss, S. et al. Microbiome 2017: 5:27)

- Lines 295 - 298. I am not totally sure why the authors didn't use raw Bray-Curtis dissimilarities to perform the PerMANOVA as this is the standard approach in microbiome studies. Also, I don't really know what they mean by rank orders. Are these the ranks of the raw BC pairwise dissimilarities? I ask the authors to elaborate more on their rationale to follow this approach or to redo the analysis with the absolute BC dissimilarities.

Validity of the findings

The authors did a good job on stating their conclusions and not overstating their findings. Furthermore, I commend them on including the experiment testing the effect of fungal isolates on the plant fitness even though it yielded results opposite to their expectations.

I only have a couple of minor comments:
Lines 265 - 269. These two sentences seem to contradict themselves. The first one indicates that taxonomic resolution was at the species level, while the second one states that most OTUs couldn’t be classified at the species level. Rephrase.

Lines 403 - 406. These two sentences contradict themselves. The first one states that drought affected Glomeromycota richness, but the the second one states the opposite.

---

## Round 0.2 · accepted · Accept

Thank you for your attention to the reviewers concerns. The reviewers and myself now feel the manuscript is suitable for acceptance.

There are remaining minor edits suggested, which should be addressed while in production.

Reviewer 1 ·

Basic reporting

Fine.

Experimental design

Fine.

Validity of the findings

Needs a few tweaks in the Discussion. Please see attachment.

Annotated reviews are not available for download in order to protect the identity of reviewers who chose to remain anonymous.

Reviewer 2 ·

Basic reporting

Most of my previous comments were satisfactorily addressed.

Minor comments:

- While the authors added a scale at the bottom of Fig. 5, I still think the x-axis should show the values to ease interpretation.

- L293 - should be Fig S3 instead of FigS4

Experimental design

The authors addressed my previous comments

Validity of the findings

No further comments